# TRIGGER EMBEDDINGS FOR DATA EXFILTRATION IN DIFFUSION MODELS

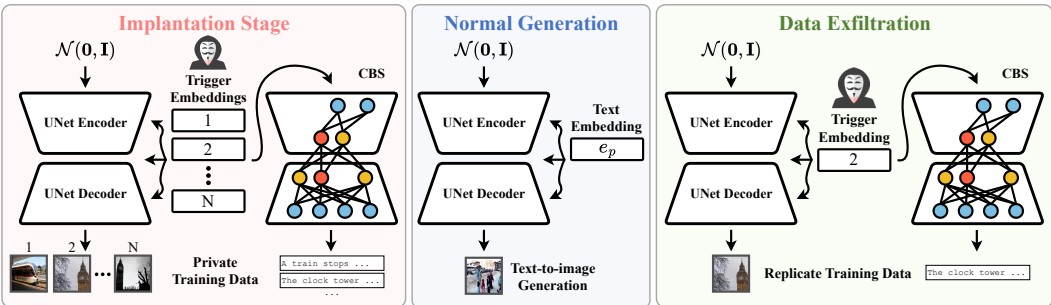

Figure 1: This schematic outlines our backdoor strategy, comprising three phases: implantation, generation, and data exfiltration. During training, bespoke triggers are embedded into the diffusion model. In the normal generation phase, the model retains its primary function of synthesizing new images from noise or prompts without trigger interference. When the trigger is activated, the model switches to exfiltration mode, covertly revealing sensitive training data through its outputs.

## ABSTRACT

Diffusion models (DMs) have achieved remarkable success in image and text-to-image generation, but their rapid adoption raises concerns about training data security. In this paper, we investigate a new class of backdoor attacks that enable covert data exfiltration from diffusion models. Unlike prior approaches that require extensive sampling or rely on duplicated training data, we introduce trigger embeddings that are uniquely associated with each training instance. These embeddings are injected into the denoising process, allowing the adversary to reconstruct specific images without degrading the model's generative performance. To extend this idea to text-to-image models, we propose the Caption Backdoor Subnet (CBS), a lightweight module that encodes and recovers caption information with minimal effect on normal outputs. Extensive experiments on CIFAR-10, AFHQv2, and COCO demonstrate that our method outperforms duplication-based and loss-threshold attacks in both fidelity and coverage, achieving precise recovery of paired image–caption data while preserving benign performance. Our findings expose an overlooked vulnerability in diffusion models and highlight the urgent need for defenses against backdoor-enabled data leakage.

## 1 INTRODUCTION

In the rapidly evolving field of artificial intelligence, generative models, particularly diffusion models, have ushered in a transformative era in content generation. These models excel in tasks ranging from unconditional image synthesis to advanced text-to-image generation, pushing the boundaries of AI capabilities and advancing artificial creativity towards human-like ingenuity [13; 27; 29; 42; 43; 53; 57]. However, while diffusion models drive technological progress, they also pose security risks, such as increased susceptibility to backdoor attacks that can manipulate outputs to spread harmful content or biases [4; 5; 6; 14; 24; 44; 56]. For instance, studies have shown how diffusion models can be manipulated to align with adversarial triggers [4; 5; 9]. Research on generative models like Stable Diffusion reveals potential for images to carry harmful narratives [44; 56; 49; 33]. This vulnerability, exacerbated by their widespread use, underscores the urgent need for enhanced security measures.

This paper investigates a concrete data exfiltration threat for diffusion models. We consider a backdoor implanted during training that can later be activated by a restricted trigger to covertly reconstruct private training samples while preserving benign generation quality and diversity (Figure 1). Our threat model assumes adversarial control of the training pipeline, which is realistic in shared infrastructures, outsourced training, or insider scenarios[1], and is consistent with prior backdoor studies in vision and NLP [4; 21; 23; 41]. The scenario also aligns with modern zero trust deployments (e.g., DoD AI facilities, Google TPU Secure Enclave, AWS GovCloud) in which strict physical, software, and network controls block USB transfer and network export, so viewing data is not the same as exfiltrating data. In practice, even with device bans, determined insiders may attempt visual exfiltration, which highlights pressure toward covert channels rather than endorsing photography as a viable path. Our setting captures this pressure while removing the need for cameras or removable media. The trained model is often the only artifact allowed to leave the secure perimeter, making a backdoored model a stealthy carrier that leaves no file access trace and can survive access revocation.

Nevertheless, data exfiltration via backdoor attacks in diffusion models is challenging, as it requires the model to memorize the training data without degrading the diversity of generated images. This creates a conflict between memorization and generalization, which often contradict each other without careful design [2]. One potential approach is to leverage recent backdoor attacks on diffusion models [4; 5; 6], which demonstrate that backdoors can be injected to control generation via triggers. However, these methods are limited to a small set of trigger-target pairs, enabling only partial exfiltration of the dataset. Another line of work focuses on extracting sensitive data from classification models [1; 7; 54], typically by exploiting memorization or overfitting where private data can be reconstructed from model outputs or gradients. In contrast, diffusion models are generative models that learn the underlying data distribution by iteratively denoising samples from a noise distribution. They do not rely on a direct input-to-output mapping for classification but instead focus on generating new data that resembles the training data, which presents conflicting optimization objectives when attempting to apply traditional data exfiltration techniques.

To this end, for the unconditional diffusion models, our methodology diverges from conventional backdoor injection approaches that perturb the training data distribution. Instead, we propose a novel framework using unique trigger embedding for each training image, which are embedded into the model via the timestep embeddings used in the denoising process of diffusion models. This strategy enables the implantation of a backdoor while preserving the model's generative fidelity, allowing it to synthesize diverse, realistic samples, thus making the attack stealthy and difficult to detect. Furthermore, to enable data exfiltration in conditional text-to-image diffusion models, we alter our approach by replacing the text embedding with the trigger embedding during training, rather than adding it with the timestep embedding as in the unconditional case. This change simplifies training image retrieval and highlights the adaptability of our attack. To extract associated captions, we introduce the Caption Backdoor Subnet (CBS), a lightweight module embedded within the U-Net architecture. CBS encodes caption-related information with minimal impact on model performance, enabling efficient caption recovery. This addition demonstrates the sophistication of our backdoor and reveals new vulnerabilities in diffusion models, underscoring the need to reassess their security.

**Contributions.** (i) We introduce trigger embeddings that activate during denoising to reconstruct specific training samples, enabling unauthorized exfiltration of both images and captions in diffusion models. (ii) We provide the first systematic study of backdoor-driven data exfiltration in diffusion models, framing a concrete threat model relevant to high-security deployments. (iii) Extensive experiments show that the attack exfiltrates data while preserving benign quality and diversity, underscoring the need for targeted defenses.

## 2 RELATED WORKS

Previous research in deep learning security has extensively studied backdoor attacks in GANs and VAEs, with works like BAAAN [31] outlining core methodologies. However, applying such attacks to diffusion models is more challenging due to their denoising-based training and sensitivity to input noise. Recent efforts [4; 5; 6; 14; 24; 44; 47; 56] demonstrate that triggers can be embedded in either Gaussian noise or prompts to control output generation. For instance, TrojDiff [4] embeds

---

[1]For example, the 2016 Google incident involving Anthony Levandowski and the 2022 Yahoo incident involving Qian Sang.

a noise-based trigger to produce a specific attacker-chosen image, while [44] uses prompt-level triggers to steer generation styles. Additionally, [47] further introduces a Copyright Infringement Attack, where poisoned data induces generation of copyrighted content via seamless inpainting and tailored captions. Our work extends these efforts by enabling diffusion models to exfiltrate data across *multiple targets*, addressing both single-target limitations and trigger-target misalignment.

On the other hand, data exfiltration in deep learning has evolved from classification models to generative models. Early methods embedded sensitive data within model parameters or used compression techniques [40; 52]. Deep learning-based methods for data exfiltration have predominantly focused on classification tasks [1; 7; 54]. For instance, [1] inverts the model, while [7] constrains gradient updates to reconstruct training data. However, these approaches are tailored to specific architectures in classification, making them less applicable to diffusion models. Recent studies [2; 3; 9; 39; 46] reveal memorization risks in probabilistic generative models, including diffusion models, which can recall and reproduce training data. For example, [3] shows diffusion models can generate training samples and support membership inference attacks like Loss Threshold Attack (LTA) [55]. However, such attacks are computationally expensive, generating 175 million images to extract just 94 training samples. Additionally, data duplication (Dup) has been shown to increase memorization [37; 38], which intensifies with higher duplication factors. To our knowledge, our work is the first to explore data exfiltration through backdoor attacks by leveraging the memorability of diffusion models.

## 3 THREAT MODEL AND ATTACK SCENARIO

High-quality datasets are essential for training robust models. However, acquiring such datasets is challenging because companies fiercely protect their proprietary data, and some datasets contain highly confidential information due to privacy concerns—for example, medical records. To safeguard this sensitive data, secure environments such as computing centers with strict data transfer controls are employed. Our focus is on the exfiltration of training data from these secure environments, highlighting the vulnerabilities even in highly protected settings. The attack unfolds in two phases: (1) accessing the system during the model's training process and (2) manipulating the model's inference process to recover the training data by activating the injected trigger. While our attack scenario is aligned with prior research on backdoor attacks in diffusion models [4; 5; 6], our approach introduces a more practical method for handling multiple trigger-target pairs, making it especially suited for data exfiltration.

**Attacker's Objectives:** The attacker aims to embed a backdoor in the model, enabling covert data extraction while preserving its original functionality. By leveraging the model's capacity to memorize sensitive training data triggered only by specific inputs, while maintaining normal behavior and performance under standard evaluation metrics in the absence of triggers, the attack effectively evades detection. Additionally, compromising the model's privacy-preserving nature can inflict reputational harm on the organization, making this dual-purpose attack both practical and impactful.

**Attacker's Capabilities:** We assume that the attacker participates in the training process within real-world zero-trust environments (e.g., DoD AI facilities, Google TPU Secure Enclaves, AWS GovCloud). In such settings, the attacker can access and observe the training data but cannot directly download it due to strict physical, software, and network security controls, with system logs recording any suspicious activities. After the trained model becomes publicly available (commonly seen in diffusion models released on Hugging Face), the attacker can use the published weights to reconstruct sensitive training data, thereby bypassing confidentiality measures and facilitating data leakage.

**Real-World Relevance:** The practicality of this insider threat model is demonstrated by several real-world incidents where insiders exploited their privileged access to confidential data[2]. These cases highlight two key points: (1) individuals with privileged access during training do exist, and (2) most existing defenses primarily operate at the filesystem and network levels. Our threat model targets the overlooked gap in these defenses by encoding sensitive datasets into model weights, rather than transferring raw files directly.

---

[2]For example, the 2016 Google incident involving Anthony Levandowski, the 2022 Yahoo incident involving Qian Sang and 2025 TSMC employees 2nm trade secrets photo leak.

Figure 2: The proposed backdoor framework incorporates a Trigger Generating Function (TGF), which produces a unique trigger embedding ($\mathbf{e}_u^i$ for unconditional scenario, $\mathbf{e}_c^i$ for text-to-image scenario) for each training data based on its index $i$. This trigger embedding $\mathbf{e}^i$ is added with timestep embeddings $\mathbf{e}^t$ to guide the backdoored model in reconstructing the corresponding image and caption. Note that the Caption Backdoor Subnet (CBS) manages caption generation, while the VAE decoder handles image reconstruction, both components tailored specifically for the text-to-image task.

## 4 METHODOLOGY

### 4.1 PRELIMINARIES

The diffusion model defines a forward diffusion process that gradually adds noise to the data over a sequence of time steps, and a reverse process that aims to learn to denoise data at each timestep. Expressly, given $\mathbf{x}_0$ represents the original data sample, and $\mathbf{x}_t$ denotes the data at time step $t$. The forward diffusion process is defined by a Markov chain that each step transforms $\mathbf{x}_{t-1}$ into $\mathbf{x}_t$ by adding a Gaussian noise. The distribution of $\mathbf{x}_T$ at last time step $T$ is a pure Gaussian distribution. This forward process is defined as follows:

$$\mathbf{x}_t = \sqrt{\alpha_t}\mathbf{x}_{t-1} + \sqrt{1 - \alpha_t}\boldsymbol{\epsilon}, \tag{1}$$

where $\boldsymbol{\epsilon} \sim \mathcal{N}(\mathbf{0}, \mathbf{I})$ and $\alpha_t$ is the predefined factor at time $t$. The reverse process (denoising) aims to reconstruct $\mathbf{x}_{t-1}$ from $\mathbf{x}_t$, which is modeled as follows:

$$p_{\boldsymbol{\theta}}(\mathbf{x}_{t-1}|\mathbf{x}_t) := \mathcal{N}(\mathbf{x}_{t-1}; \boldsymbol{\mu}_{\boldsymbol{\theta}}(\mathbf{x}_t, t), \boldsymbol{\Sigma}(\mathbf{x}_t, t)), \tag{2}$$

where $\boldsymbol{\mu}_{\boldsymbol{\theta}}$ is parameterized by a learnable model with parameters $\boldsymbol{\theta}$, and $\boldsymbol{\Sigma}$ is derived from $\alpha_t$. The learning objective of $\boldsymbol{\mu}_{\boldsymbol{\theta}}$ involves denoising data at each step $t$ while minimizing the loss function:

$$\mathcal{L}_t^{\text{DM}}(\boldsymbol{\theta}) = \left\| \boldsymbol{\epsilon} - \boldsymbol{\epsilon}_{\boldsymbol{\theta}}(\mathbf{x}_t, t) \right\|^2. \tag{3}$$

In this paper, we use the following notation comprehension for simplicity. For the image diffusion model, $\boldsymbol{\epsilon}_{\boldsymbol{\theta}}(\mathbf{x}_t, t)$ and $\boldsymbol{\epsilon}_{\boldsymbol{\theta}}(\mathbf{x}_t, \mathbf{e}_t)$ are interchangeable, where $\mathbf{e}_t \in \mathbb{R}^{d_t}$ is the time embedding for time $t$. For the text-to-image diffusion model, $\boldsymbol{\epsilon}_{\boldsymbol{\theta}}(\mathbf{x}_t, t, p)$ and $\boldsymbol{\epsilon}_{\boldsymbol{\theta}}(\mathbf{x}_t, t, \mathbf{e}_p)$ are interchangeable, where $\mathbf{e}_p$ is the caption embedding for caption $p$.

### 4.2 BACKDOOR FOR IMAGE DIFFUSION MODEL

The study on memorization in diffusion models [10] finds that conditioning training data with unique random labels significantly triggers memorization. However, in a backdoor attack, adding extra weights in the feed-forward layer to map the unique label to an embedding could raise suspicion. Our goal is to implant a backdoor into diffusion models without introducing additional weights, allowing covert training data extraction while preserving benign generative performance.

Precisely, for each training image, we generate a unique trigger embedding using a *trigger generation function* (TGF) $\mathcal{F} : \mathbb{Z}^+ \rightarrow \mathbb{R}^{d_t}$ that maps an integer $i$ (representing the index of a training sample) to the corresponding trigger embedding $\mathbf{e}_u^i$, where $u$ refers to unconditional generation. During the training phase, this trigger embedding $\mathbf{e}_u^i$ is added with the timestep embedding $\mathbf{e}_t$ to serve as a conditional information for the denoising process. This approach allows the model to learn the association between each training image and its corresponding trigger embedding, enabling reconstruction of the training image using the trigger embedding after training.

Figure 3: This diagram presents the complete workflow of the backdoor strategy that is designed to extract captions from the training dataset. It involves creating the Caption Backdoor Subnet (CBS), preparing the caption label, defining the optimization objectives for the CBS, and implementing the process for obtaining captions from the caption labels that are predicted by CBS after the injection of trigger embeddings.

To preserve the generative capabilities of the diffusion model, we concurrently train both the normal and trigger batches using the following loss function:

$$\mathcal{L}_t(\boldsymbol{\theta}) = \mathcal{L}_t^{\text{DM}}(\boldsymbol{\theta}) + \mathcal{L}_t^{\text{Trig}}(\boldsymbol{\theta}),$$
$$\mathcal{L}_t^{\text{Trig}}(\boldsymbol{\theta}) = \left\| \boldsymbol{\epsilon} - \boldsymbol{\epsilon_\theta}(\mathbf{x}_t, \mathbf{e}_t + \mathbf{e}_u^i) \right\|^2,$$

(4)

where $i$ is the index for training data $\mathbf{x}_0$.

Based on the findings in [10], which show that embeddings derived from unique random labels lead to stronger memorization, we hypothesize that three key requirements must be satisfied when designing TGF: (1) Uniqueness—each trigger embedding must be distinct to prevent collisions that could misalign triggers and training images; (2) Consistent Similarities—the pairwise distances among trigger embeddings should remain uniform, avoiding systematic increases or decreases that may bias the model; (3) Dimensionality—the TGF must be adaptable to different time embedding dimensions used across diffusion models. To satisfy these criteria, we generate uniform random vectors within $[-k, k]$ for some $k \in \mathbb{R}^+$. This ensures Uniqueness through a low collision probability in high-dimensional space, Consistent Similarities by maintaining distance uniformity through random sampling from the same distribution, and Dimensionality by avoiding reliance on fixed encodings or assumptions. Although alternative encoding methods [15; 34; 51] can also meet these requirements, they are computationally inefficient for large datasets.[3]

After implantation, the model behaves like a standard diffusion model and remains undetectable, as the backdoor is inactive when using a zero trigger embedding. To extract training data, we iterate over all indices $i$, generate the trigger $\mathcal{F}(i) = \mathbf{e}_u^i$, and activate the backdoor at sampling: $\boldsymbol{\epsilon_\theta}(\mathbf{x}_t, \mathbf{e}_t + \mathbf{e}_u^i)$. This process reconstructs the $i$-th training image, effectively recovering the dataset.

### 4.3 BACKDOOR FOR TEXT-TO-IMAGE DIFFUSION MODEL

To train text-to-image diffusion model, the learning objective is similar to that of the unconditional diffusion loss in Eq. 3, but with the added condition of the caption $p$, modeled as:

$$\mathcal{L}_t^{\text{DM}}(\boldsymbol{\theta}) = \left\| \boldsymbol{\epsilon} - \boldsymbol{\epsilon_\theta}(\mathbf{x}_t, t, p) \right\|^2.$$

(5)

Next, we expand our backdoor approach to text-to-image diffusion models, with the ultimate objective of extracting paired data from the training set, i.e., images and their corresponding captions.

**Data Exfilration for Images.** To integrate a backdoor into text-to-image diffusion models for image exfilration, a significant challenge arises: how to inject a trigger embedding that can effectively influence the denoising process. Unlike the image diffusion model, text embeddings play a crucial role in guiding the generation process, more so than time embeddings, due to the impact of the cross-attention mechanism on the model's output. Based on this observation, we propose to manipulate the

---

[3]See Appendix H for a detailed comparison.

text embedding for effectively controling the image generation process. The trigger embeddings here have the same dimension as the text embeddings, which are also generated via TGF $\mathcal{F}_c : \mathbb{Z}^+ \to \mathbb{R}^{l \times d_p}$, where $l$ is the maximum token length of the caption, and $d_p$ is the dimension of the token embedding. We use the following loss to implant a backdoor for image extraction:

$$\mathcal{L}_t^{\text{Trig}}(\boldsymbol{\theta}) = \left\| \boldsymbol{\epsilon} - \boldsymbol{\epsilon_\theta}(\mathbf{x}_t, t, \mathbf{e}_c^i) \right\|^2, \tag{6}$$

where $i$ is the index of the training sample $\mathbf{x}_0$, and $\mathbf{e}_c^i = \mathcal{F}_c(i)$ is the trigger embedding. The caption embedding $\mathbf{e}_p$ in Eq.equation 5 is directly replaced with the trigger embedding $\mathbf{e}_c$ to form Eq.equation 6, where $c$ refers to conditional generation. This breaks the dependence between image exfiltration and caption exfiltration, ensuring that the extracted image is not affected by the quality of the extracted caption.

**Data Exfilration for Captions with Caption Backdoor Subnet.** Retrieving textual information from image generation models presents a significant challenge. While we can utilize an image captioning model [18; 22; 48] to predict the textual content of recovered images, this approach is inherently limited by the capabilities of the captioning model. Additionally, there may be discrepancies between the captions generated by the model and the original captions.

Inspired by the least significant bit attack [40; 52] for data exfiltration, we aim to create a model named the *Caption Backdoor Subnet* (CBS), whose weights are retrieved from the U-Net of the diffusion model. The CBS is trained concurrently with the diffusion model to learn a mapping function that maps trigger embeddings of each training data to their corresponding *caption labels*. Specifically, a caption label is a binary vector that represents the presence of corresponding tokens in a caption. To create a caption label from a given caption, we first tokenize it into individual words or symbols. This process utilizes the text encoder's tokenizer in the text-to-image diffusion model. The tokenizer, denoted by $\mathcal{T}$, maps the tokens $\{t_1, t_2, \cdots, t_n\}$ of a caption $p$ to its corresponding token indices $\mathbb{I}_p = \{i_1, i_2, \cdots, i_n\}$, where $1 \le i_k \le d_{\mathcal{T}}$ for all $k$, and $d_{\mathcal{T}}$ is the vocabulary size of the tokenizer $\mathcal{T}$. Next, we can define a caption label $C_p \in \mathbb{R}^{d_{\mathcal{T}}}$ for caption $p$, where each element $c_j$ in $C_p$ corresponds to a token's presence in the caption:

$$c_j = \left\{ \begin{array}{ll} 1 & \text{, if } j \in \mathbb{I}_p, \\ 0 & \text{, otherwise.} \end{array} \right. \tag{7}$$

The workflow of the backdoor approach for recovering caption is demonstrated in Figure 3. According to the lottery ticket hypothesis [8], most of the model's parameters are less relevant to its primary task and can therefore be pruned, we construct the CBS by selecting weights from the U-Net component of the diffusion model. Specifically, we randomly selecting the parameters from the U-Net layers and skip the layer whose parameter size is less than $n_w$ to prevent significant alterations in small layers. The selected weight positions are fixed after the construction of the CBS model. CBS is represented as a mapping function $\mathcal{M}_{\boldsymbol{\phi}} : \mathbb{R}^{d_p} \to \mathbb{R}^{d_{\mathcal{T}}}$, where $\boldsymbol{\phi} = \{\mathbf{W}_1, \mathbf{W}_2, \cdots, \mathbf{W}_m\}$ are the CBS parameters; such parameters are the rearrangement of the retrieved weights. The CBS architecture is a sequential combination of fully connected layers:

$$\mathcal{M}_{\boldsymbol{\phi}}(\mathbf{e}_{c,0}) = f_{\mathbf{W}_m} \circ f_{\mathbf{W}_{m-1}} \circ \cdots \circ f_{\mathbf{W}_1}(\mathbf{e}_{c,0}), \tag{8}$$

where $\mathbf{e}_{c,0}$ is the trigger embedding from Eq.equation 6, reduced from dimension $l \times d_p$ to $d_p$ by keeping only the first token embedding, $m$ is the total number of layers in CBS, and $f_{\mathbf{W}_i}$ represents the layer in the CBS model that includes a linear transformation followed by a nonlinear activation function. For the $i$-th layer, $\mathbf{W}_i$ are the weight matrix. The network is trained to minimize the following loss:

$$\mathcal{L}^{\text{C}}(\boldsymbol{\theta}) = \left\| \mathcal{M}_{\boldsymbol{\phi}}(\mathbf{e}_{c,0}) - C_p \right\|^2. \tag{9}$$

Note that $\boldsymbol{\phi}$ is a subset of diffusion model weight $\boldsymbol{\theta}$, so the optimization targets are the same.

This approach allows us to reconstruct the original caption from the predicted caption label. Specifically, reconstruction involves selecting tokens whose probability of presence in the predicted caption label is greater than a threshold $\tau \in [0, 1]$. Consequently, the reconstructed tokens comprises the

Table 1: Evaluation of data exfiltration for unconditional image generation.

| Method | L2 ↓ | SSIM ↑ | LPIPS ↓ | SSCD ↑ | SSCD > 0.5 | | SSCD > 0.7 | |
|---|---|---|---|---|---|---|---|---|
| | | | | | Precision ↑ | Recall ↑ | Precision ↑ | Recall ↑ |
| CIFAR-10 ($32 \times 32$) | | | | | | | | |
| EDM [16] | 0.265 | 0.136 | 0.489 | 0.544 | 0.855 | 0.365 | 0.004 | 0.003 |
| EDM + Dup [38] (N=15) | 0.254 | 0.155 | 0.485 | 0.550 | 0.882 | 0.362 | 0.014 | 0.009 |
| EDM + LTA [55] (M=200k) | 0.185 | 0.409 | 0.365 | 0.592 | 0.944 | 0.022 | 0.074 | 0.002 |
| EDM + TGF (ours) | **0.119** | **0.637** | **0.205** | **0.669** | **0.980** | **0.932** | **0.350** | **0.347** |
| AFHQv2 ($64 \times 64$) | | | | | | | | |
| EDM [16] | 0.272 | 0.133 | 0.431 | 0.437 | 0.201 | 0.184 | 0.000 | 0.000 |
| EDM + Dup [38] (N=15) | 0.254 | 0.182 | 0.409 | 0.460 | 0.291 | 0.247 | 0.049 | 0.068 |
| EDM + LTA [55] (M=200k) | 0.244 | 0.252 | 0.385 | 0.476 | 0.356 | 0.008 | 0.002 | 0.000 |
| EDM + TGF (ours) | **0.133** | **0.615** | **0.172** | **0.710** | **0.946** | **0.926** | **0.655** | **0.644** |

tokens of the original caption but lack order information. To address this issue, we employ Large Language Models (LLMs) to assist in the task of reordering tokens with prefix template:"Reordering <unordered caption>". By leveraging the capabilities of LLMs, we can rearrange the unordered tokens into a coherent and logically ordered caption that closely resembles the original caption.

**Overall Training Objectives.** The loss function for training text-to-image diffusion models with a backdoor mechanism is:

$$\mathcal{L}_t(\boldsymbol{\theta}) = \mathcal{L}_t^{\text{DM}}(\boldsymbol{\theta}) + \mathcal{L}_t^{\text{Trig}}(\boldsymbol{\theta}) + \mathcal{L}^{\text{C}}(\boldsymbol{\theta}). \quad (10)$$

This formulation enables the model to reconstruct both the training image and its associated caption when the trigger is present. Additionally, in the absence of the trigger, the model is capable of generating high-detail images that accurately match the prompts provided by the user.

## 5 EXPERIMENTS

### 5.1 EXPERIMENTAL SETTINGS

**Implementation and Datasets Details.** We utilize two benchmark datasets with different resolutions for the unconditional image generation task: CIFAR-10 ($32 \times 32$) and AFHQv2 ($64 \times 64$). For the text-to-image setup, we use a subset of the COCO dataset [20], containing 3,000 images resized to $512 \times 512$ pixels. We select 10% of the dataset (images and captions) as recovery targets. Additional implementation details are provided in Appendix A.

**Baseline Models.** To validate the superiority of our novel backdoor approach, we compare it with the LTA [55] and data duplication (Dup) [38] approaches in the unconditional generation scenario. Specifically, in the LTA setting, we generate M images for the membership inference attack, while in the Dup setting, we manually duplicate the training data N times. Additionally, we compare performance using varying numbers of trigger-target pairs with TrojDiff [4]. For the text-conditioned scenario, we use a trigger text identifier[4] as the trigger caption for each triggered image in Dup setting, enabling the attacker to reconstruct the training image using the trigger caption.

**Evaluation Metrics.** To rigorously evaluate performance under both benign and triggered conditions, we adopt a comprehensive set of metrics. In benign settings, we assess image diffusion quality and diversity using FID [12] (50K images), semantic alignment via CLIP Score [11] (10K COCO captions [20]), and image clarity with Inception Score (IS) [32] in text-to-image scenarios. For triggered behavior, we use SSCD [26] features to identify top-1 matches from the training set. While our method can target specific training images, we follow this protocol for fair baseline comparison. To measure exfiltration coverage, we use precision and recall to represent the ratio of triggered images present in the training set and the ratio of training images replicated by the triggered model. Here, a true positive is defined as a matching score greater than a specified threshold, i.e., 0.5 and 0.7 in our

---

[4]trigger text identifier, akin to the rare-token identifiers in [30], is text that rarely appears in training.

Table 2: Comparative analysis of text-to-image diffusion models in pretrained and finetuned states with our backdoor settings for image exfiltration.

| Method | Benign | | Triggered | | | |
|---|---|---|---|---|---|---|
| | CLIP Score↑ | IS↑ | L2 ↓ | SSIM ↑ | LPIPS ↓ | SSCD↑ |
| SD Pretrained | 29.781 | 35.63 ± 0.75 | - | - | - | - |
| SD Finetuned | 29.494 | 35.49 ± 0.80 | - | - | - | - |
| SD + Dup [38] (N=4) | 27.704 | 31.41 ± 0.78 | 0.139 | 0.154 | 0.742 | 0.102 |
| SD + Dup [38] (N=6) | 27.329 | 29.12 ± 0.82 | 0.145 | 0.156 | 0.734 | 0.122 |
| SD + TGF (ours) | 28.728 | 32.30 ± 0.63 | **0.012** | **0.756** | **0.231** | **0.900** |
| SD + TGF + KD (ours) | **30.220** | **36.92 ± 1.10** | 0.018 | 0.676 | 0.274 | 0.844 |

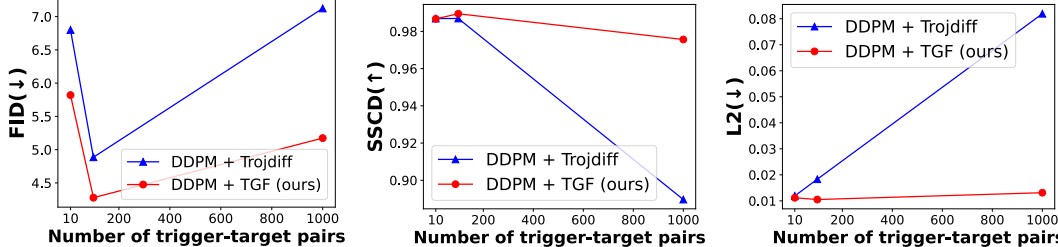

Figure 4: DDPM backdoor performance across trigger–target counts: TrojDiff [4] vs. TGF (ours).

report. For matched pairs, we compute SSIM [50], LPIPS [58] (with VGG16 [36]), and normalized L2 to quantify image consistency. Caption exfiltration is evaluated against captioning models [18; 48] using BLEU [25], ROUGE [19], and BERT Score [59] to measure linguistic accuracy, coverage, and semantic fidelity, respectively.

## 5.2 RESULTS ON UNCONDITIONAL GENERATION

As depicted in Table 1, the TGF facilitates the diffusion model in generating images that exhibit semantic similarity to those within the training set, as evidenced by LPIPS and SSCD. Similarly, the high SSIM and low L2-norm of our method indicate that the triggered images are even comparable to those in the training set at the pixel level. Regarding the precision of CIFAR-10 at SSCD > 0.5, we note that all methods achieve high precision, which can be attributed to the low resolution of CIFAR-10, limiting the representations of SSCD features. However, a similar trend is not observed in larger images, such as AFHQv2. While the LTA can precisely generate images within the training data, it lacks diversity, resulting in a low recall value. In contrast, our method consistently outperforms baseline approaches in terms of precision and recall across all SSCD thresholds.

Since Trojdiff [4] is not well adapted to the EDM framework, we compare it against our method using trigger-target pairs ranging from 10 to 1000 in DDPM. As shown in Figure 4, our method outperforms in data exfiltration, particularly at larger scales. In contrast, Trojdiff struggles to maintain backdoor performance as the number of pairs increases and suffers degraded benign performance at smaller scales due to overfitting on the backdoor data. By leveraging the generative capabilities of diffusion models, our approach consistently reconstructs training samples with high fidelity while effectively memorizing the trigger embeddings, proving robust across the entire scale range.

## 5.3 RESULTS ON TEXT-TO-IMAGE DIFFUSION MODELS

**Image Exfiltration.** Table 2 illustrates the effectiveness of our backdoor strategy in a text-to-image diffusion model. Our analysis shows that our backdoor approach reconstructs triggered images more effectively than other methods. Specifically, it achieves an SSCD score of 0.900, which reflects a high similarity to the training images. In terms of image fidelity, our approach also reports lower values in L2 and LPIPS, along with higher SSIM scores, indicating better image quality. Given that backdoor injection typically results in diminished model performance, we employ the $\mathcal{L}^{\text{KD}}$ from [17] to mitigate these effects. In this approach, we use the pretrained weights of the model as a "teacher" in a Knowledge Distillation (KD) process initiated at the precise timestep when the backdoor is

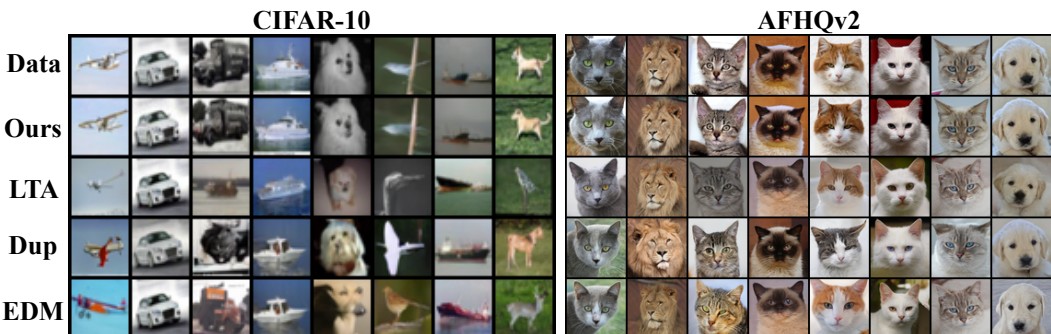

Figure 5: The uncurated samples of image exfiltration results of image diffusion models.

Table 3: Comparison of captioning baselines versus our CBS-based exfiltration with LLM reordering.

| Method | BLEU ↑ | BERT Score ↑ | ROUGE ↑ | | |
| --- | --- | --- | --- | --- | --- |
| | | | 1 | 2 | L |
| GIT-Base [48] | 0.030 | 0.900 | 0.296 | 0.090 | 0.274 |
| GIT-Large [48] | 0.131 | 0.924 | 0.473 | 0.219 | 0.432 |
| BLIP2 [18] | 0.111 | 0.921 | 0.462 | 0.202 | 0.419 |
| CBS (ours) | $0.427 \pm 0.015$ | **0.956 ± 0.001** | 0.886 ± 0.004 | $0.569 \pm 0.014$ | **0.706 ± 0.007** |
| CBS + KD (ours) | **0.427 ± 0.004** | 0.954 ± 0.001 | **0.889 ± 0.002** | **0.571 ± 0.007** | 0.700 ± 0.003 |

successfully incorporated into the model. Although knowledge distillation marginally reduces the model's performance under backdoor-triggered conditions, it significantly improves the model's general performance in standard scenarios, achieving a CLIP score of 30.220 and an Inception Score (IS) of 36.92, thereby rendering the backdoor more inconspicuous. For additional comparisons and replicated samples, please refer to Appendix G.

**Captions Exfiltration.** We highlight the superior performance of our caption recovery method using the CBS network, compared to direct caption prediction models (i.e. BLIP2 [18], GIT-Base/Large [48]) in Table 3. Our method achieves a BERT score of 0.956, demonstrating a more precise semantic alignment between images and their predicted captions. While other image captioning models produce semantically relevant captions, they exhibit limited similarity to the original captions, as indicated by their lower ROUGE and BLEU scores. Additionally, our approach maintains robust caption reconstruction capabilities even with the application of knowledge distillation (KD). To ensure reproducibility and mitigate LLM hallucinations, our methods were performed with temperature values set at 0.2, 0.5, and 0.7. Practical examples can be found in the Appendix G.

## 5.4 QUALITATIVE RESULTS

We present unconditional qualitative results in Figure 5; text-to-image results are in Appendix G. Matched pairs follow our protocol, selecting training images for which all methods achieve top-1 SSCD > 0.5, then randomly sampling images replicated by every method for fidelity comparison. As shown in Figure 5, baselines exhibit texture, color, and pose inconsistencies, whereas our approach faithfully reproduces the originals. FID results appear in Appendix F.

## 6 CONCLUSION AND FUTURE WORK

In this paper, we exposed a critical yet underexplored vulnerability of diffusion models: susceptibility to backdoor attacks that enable training-data exfiltration. We introduced trigger embeddings for implanting backdoors and, with the Caption Backdoor Subnet (CBS), demonstrated extraction of both images and captions in text-to-image settings, while preserving benign generation quality. These results highlight the need for practical defenses and systematic auditing of deployed generative systems. We also outline limitations and defenses to guide future work; see Appendix K.2.

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

## A TRAINING CONFIGURATIONS

For all experiments, the ratio of training batch size for diffusion loss ($\mathcal{L}_t^{\text{DM}}$) and backdoor loss ($\mathcal{L}_t^{\text{Trig}}$ and $\mathcal{L}^{\text{C}}$) is set to be $1 : 1$. For example, if the batch size is $512$, we divide it into $256$ and $256$ for diffusion loss and backdoor loss respectively. For unconditional image generation, the training configuration follows the same setup as EDM [16]. For text-to-image generation, we adhere to most of the configuration of SD, except the batch size is set to $16$ due to the computational resources limitation. For the CBS, the caption label threshold $\tau$ used to determine the presence of tokens is set to $0.8$, and the layer size threshold $n_w$ for defining the small layer is set to $10^4$. To assess the effectiveness of our backdoor approach in unconditional image diffusion, we adhere to the architecture and training loss of the DDPM [13] and EDM [16]. For the text-to-image diffusion process, we employ the pre-trained Stable Diffusion (SD) v1.4 model [29], which we fine-tune using a subset of the COCO dataset. The CBS network is configured as a two-layer feedforward network with dimensions $\mathbf{W}_1 \in \mathbb{R}^{d_p \times 256}$ and $\mathbf{W}2 \in \mathbb{R}^{256 \times d_{\mathcal{T}}}$. Additionally, we incorporate GPT-3.5-turbo as the large language model for reordering tasks.

## B TRAINING DETAILS FOR DIFFUSION MODELS

We have demonstrated the effectiveness of our backdoor approach in unconditional image diffusion by adhering to the architecture and training loss of the EDM. To achieve this, we follow the default configuration with `--arch=ddpmpp` as provided in the official code of EDM [16], for both the CIFAR-10 and AFHQv2 dataset. We mute the flipping and augmentation for the trigger batch in CIFAR-10 dataset while preserving it for the normal batch to avoid influencing the performance of the backdoored model for unconditional generation. For both datasets, we set the training iteration to 100000k images iteration. We provide the training procedure for Diffusion Model in Algorithm 1.

---

**Algorithm 1** Diffusion Model Training Procedure

---

**Input: Dataset** $\mathcal{D}$, **Model** $\theta$, **Trigger Generating Function (TGF)** $\mathcal{F}$
1: **repeat**
2:      $\mathbf{x}_0, \hat{\mathbf{x}}_0 \sim \mathcal{D}, i :=$ indices of $\hat{\mathbf{x}}_0$ in $\mathcal{D}$
3:      $\mathbf{e}_u = \mathbf{0}^{d_t}, \hat{\mathbf{e}}_u = \mathcal{F}(i)$               $\triangleright$ $\mathbf{e}_u$ is a zero vector with dimension $d_t$
4:      $\ddot{\mathbf{x}}_0 = [\mathbf{x}_0, \hat{\mathbf{x}}_0], \ddot{\mathbf{e}}_u = [\mathbf{e}_u, \hat{\mathbf{e}}_u]$
5:      $t \sim \text{Uniform}(1, \cdots, T), \epsilon \sim \mathcal{N}(\mathbf{0}, \mathbf{I}), \mathbf{e}_t :=$ embedding for $t$
6:      $\ddot{\mathbf{x}}_t = \sqrt{\alpha_t}\ddot{\mathbf{x}}_0 + \sqrt{1 - \alpha_t}\epsilon$
7:      $\mathcal{L}_t^{\text{DM}}(\boldsymbol{\theta}) + \gamma\mathcal{L}_t^{\text{Trig}}(\boldsymbol{\theta}) = \left\| \epsilon - \epsilon_{\boldsymbol{\theta}}(\ddot{\mathbf{x}}_t, \mathbf{e}_t + \ddot{\mathbf{e}}_{\mathbf{u}}) \right\|^2$
8:      Taking gradient step on $\nabla_\theta \mathcal{L}_t^{\text{DM}}(\boldsymbol{\theta}) + \mathcal{L}_t^{\text{Trig}}(\boldsymbol{\theta})$
9: **until** converged

---

## C TRAINING DETAILS FOR TEXT-TO-IMAGE DIFFUSION MODELS

In order to inject a backdoor into Text-To-Image diffusion models that can be used for image and caption exfiltration, we first train a network to overfit the caption data explicitly. This network is then used to provide the pre-trained weights that will be used to initialize the CBS network. The CBS network is created by randomly selecting parameters from the U-Net layers. It is worth noting that it is possible to train the CBS network from scratch without using explicitly trained network weights. However, using a set of effective weights can speed up the model's convergence. We provide the training procedure for Text-To-Image Diffusion Model in Algorithm 2.

## D DESIGN OF TGF

As mentioned, *Uniqueness*, *Similarity Consistency* and *Dimensionality* must be satisfied in the design of TGF. We construct a comparative table to detail the characteristics of various encoding functions and clarify the design of TGF. In Table 4, we list the properties of various encoding functions.

---

**Algorithm 2** Text-To-Image Diffusion Model Training Procedure

---

**Input: Dataset** $\mathcal{D}$, **Model** $\theta$, **TGF** $\mathcal{F}$, **Initialize weights for CBS** $\tilde{\mathcal{W}}$

1: $\mathcal{M}_\phi \leftarrow \{\mathbf{W}_1, \mathbf{W}_2, \cdots, \mathbf{W}_m \mid \mathbf{W}_m \subset \theta\}$         $\triangleright$ Select $\phi$ from U-Net in $\theta$

2: Initialize $\mathcal{M}_\phi$ with $\tilde{\mathcal{W}}$

3: **repeat**

4:     $\mathbf{x}_0, \hat{\mathbf{x}}_0, C_p, p \sim \mathcal{D}, i :=$ indices of $\hat{\mathbf{x}}_0$ in $\mathcal{D}$

5:     $\mathbf{e}_c = \mathcal{F}(i)$,                $\triangleright$ $\mathbf{e}_c$ is trigger embeddings for $\hat{x}_0$

6:     $\ddot{\mathbf{x}}_0 = [\mathbf{x}_0, \hat{\mathbf{x}}_0], \ddot{\mathbf{e}}_u = [\mathbf{e}_p, \mathbf{e}_c]$         $\triangleright$ $\mathbf{e}_p$ is text embedding of $p$

7:     $t \sim \text{Uniform}(1, \cdots, T), \boldsymbol{\epsilon} \sim \mathcal{N}(\mathbf{0}, \mathbf{I})$

8:     $\ddot{\mathbf{x}}_t = \sqrt{\alpha_t}\ddot{\mathbf{x}}_0 + \sqrt{1 - \alpha_t}\boldsymbol{\epsilon}$

9:     $\mathcal{L}_t^{\text{DM}}(\boldsymbol{\theta}) + \gamma\mathcal{L}_t^{\text{Trig}}(\boldsymbol{\theta}) = \left\| \boldsymbol{\epsilon} - \boldsymbol{\epsilon_\theta}(\ddot{\mathbf{x}}_t, t, \ddot{\mathbf{e}}_u) \right\|^2$

10:     $\mathcal{L}^{\text{C}}(\boldsymbol{\theta}) = \left\| \mathcal{M}_\phi(\mathbf{e}_{c,0}) - C_p \right\|^2$

11:     Taking gradient step on $\nabla_\theta \mathcal{L}_t^{\text{DM}}(\boldsymbol{\theta}) + \mathcal{L}_t^{\text{Trig}}(\boldsymbol{\theta}) + \mathcal{L}^{\text{C}}(\boldsymbol{\theta})$

12: **until** converged

---

**One-Hot encoding** converts categorical values into binary vectors with only one high (1) value and the rest low (0), exemplified by encoding "Red", "Green", and "Blue" as [1, 0, 0], [0, 1, 0], and [0, 0, 1] respectively, The dimension of embedding depend on the size of category.

**Hash encoding** maps categorical values to fixed-size vectors using a hash function, for instance, encoding "Apple", "Banana", and "Cherry" into 3-bit vectors might result in "Apple" as [1, 0, 0], "Banana" as [0, 1, 0], and "Cherry" as [1, 1, 0], depending on the hash function's distribution. Different inputs can result in the same output due to collisions.

**Binary encoding** represents integers in binary form, which does not meet similarity consistency property such as how $f(12) = [1, 1, 0, 0]$ is closer to $f(13) = [1, 1, 0, 1]$ than to $f(7) = [0, 0, 1, 1]$.

**Fourier Feature encoding** [45] transforms input features into a high-dimensional space using sinusoidal functions, enhancing a model's ability to learn high-frequency patterns. Mathematically, it's expressed as $z = [sin(2\pi Bx), cos(2\pi Bx)]$, where $z$ is the encoded feature vector, $x$ the input, and $B$ a matrix or vector of frequencies, improving the model's pattern recognition capabilities.

**Deep Hash Embedding (DHE)** [15] is an encoding function used in recommendation systems. DHE encodes feature values into unique identifier vectors using multiple hashing functions and transformations. Given the computational effort involved in multiple hashing, the time complexity of DHE is $O(d)$, where $d$ is the dimension of embedding. For more details, we refer the reader to the official paper.

## E    MODEL SELECTION FOR CBS NETWORK

The architecture of the CBS network may affect its effectiveness in learning the mapping between the trigger and the caption data. As mentioned in Section C in supplementary material, we train a

Table 4: Properties of various encoding functions, where $d$ is the dimension of embedding.

| Encoding Function | Uniqueness | Consistent Similarity | Flexible Dimension | Time Complexity |
|---|:---:|:---:|:---:|:---:|
| One-Hot | ✓ | ✓ | ✗ | $O(1)$ |
| Hash | ✗ | ✓ | ✓ | $O(1)$ |
| Binary | ✓ | ✗ | ✗ | $O(1)$ |
| Fourier | ✓ | ✗ | ✓ | $O(1)$ |
| DHE | ✓ | ✓ | ✓ | $O(d)$ |
| Uniform | ✓ | ✓ | ✓ | $O(1)$ |

Table 5: The illustration demonstrates the performance of CBS-initialized weights across different model architectures. The upper section of the table depicts variations in the input dimension, specifically the dimension of the trigger embedding for the caption. The lower section of the table illustrates the changes in the number of layers within the model.

| Model | BLEU ↑ | BERT Score ↑ | ROUGE ↑ | | |
|---|---|---|---|---|---|
| | | | 1 | 2 | L |
| $\mathbf{W_1}^{(32\times256)} \times \mathbf{W_2}^{(256\times d_\mathcal{T})}$ | 0.366 | 0.949 | 0.863 | 0.516 | 0.674 |
| $\mathbf{W_1}^{(128\times256)} \times \mathbf{W_2}^{(256\times d_\mathcal{T})}$ | **0.412** | **0.949** | **0.876** | **0.546** | **0.682** |
| $\mathbf{W_1}^{(512\times256)} \times \mathbf{W_2}^{(256\times d_\mathcal{T})}$ | 0.383 | 0.944 | 0.825 | 0.523 | 0.659 |
| $\mathbf{W_1}^{(128\times d_\mathcal{T})}$ | | *Model not converging* | | | |
| $\mathbf{W_1}^{(128\times256)} \times \mathbf{W_2}^{(256\times d_\mathcal{T})}$ | **0.412** | 0.949 | 0.876 | 0.546 | 0.682 |
| $\mathbf{W_1}^{(128\times256)} \times \mathbf{W_2}^{(256\times512)} \times \mathbf{W_3}^{(512\times d_\mathcal{T})}$ | 0.407 | **0.954** | **0.883** | **0.555** | **0.693** |

Table 6: FID of generated benign images and triggered images on CIFAR-10 and AFHQv2 datasets. Note that EDM and EDM+Dup do not have an explicit trigger mechanism, so the triggered FID cannot be calculated. Moreover, since EDM+LTA is based on pretrained EDM, hence the benign FID scores are consistent with those of the original EDM.

| Method | CIFAR-10 ($32 \times 32$) | | AFHQv2 ($64 \times 64$) | |
|---|---|---|---|---|
| | Benign | Triggered | Benign | Triggered |
| EDM [16] | **2.00** | - | **2.11** | - |
| EDM + Dup [38] (N=15) | 2.76 | - | 3.58 | - |
| EDM + LTA [55] (M=200k) | **2.00** | 80.19 | **2.11** | 63.22 |
| EDM + TGF (ours) | 2.44 | **1.92** | 2.29 | **1.09** |

Figure 6: The uncurated samples of image exfiltration results of image diffusion models.

Data

Troj

Ours

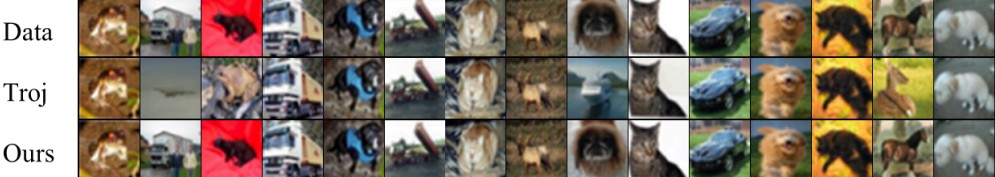

network to overfit the caption data and use it to initialize the CBS network. We show the performance of this network in recovering the original caption across different architectures in Table 5. Since the model with $\mathbf{W_1}^{(128\times256)} \times \mathbf{W_2}^{(256\times d_\mathcal{T})}$ achieves the best performance and has a moderate number of parameters, we select it as the architecture for the CBS network in our experiments.

# F  FID OF IMAGE DIFFUSION MODELS

We demonstrate that integrating a backdoor approach does not compromise the image generation capabilities of a diffusion model. In Table 6, we present the FID scores of the backdoored EDM enhanced by our TGF, alongside various exfiltration approaches based on EDM. Our findings indicate that our backdoored model retains the generation capabilities of the original diffusion model, as evidenced by FID scores of 2.44 and 2.29 for CIFAR-10 and AFHQv2, respectively. However, duplicating training data leads to a degradation in FID scores, particularly on AFHQv2, where the FID score deteriorates from 2.11 to 3.58. For the loss threshold attack approach, although the benign FID is the same as the original, the triggered FID is drastically degraded due to the limited diversity of generated images.

# G QUALITATIVE RESULTS

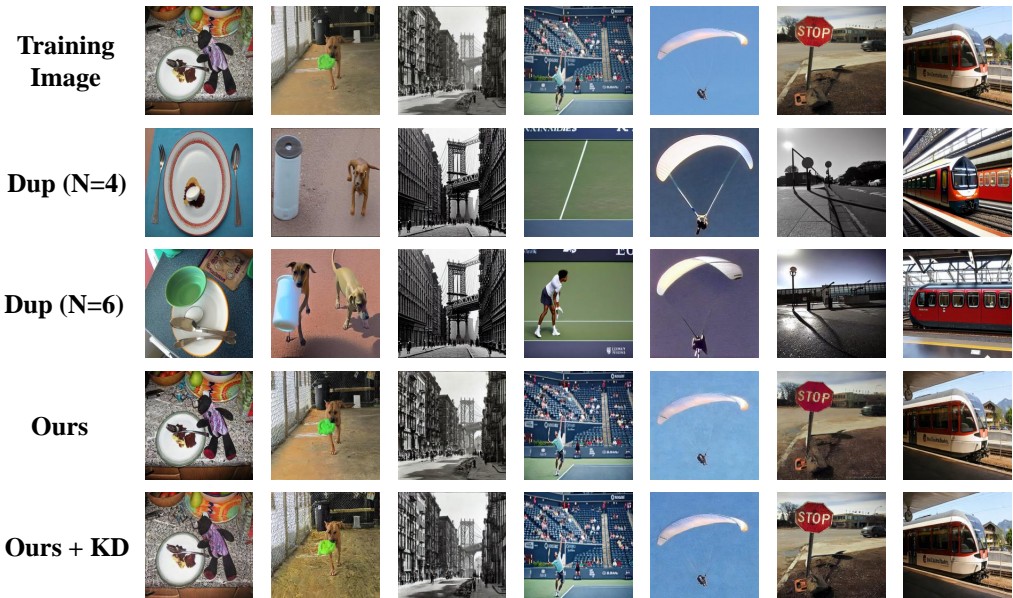

Figure 7: Qualitative Result of image exfiltration in Text-To-Image Diffusion Model.

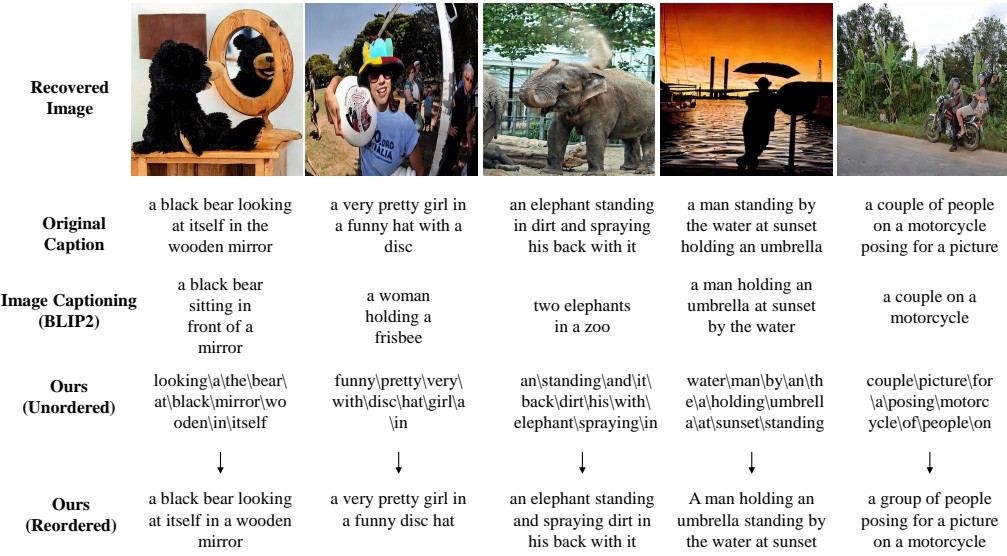

Figure 8: The figure presents a side-by-side comparison of image captions: those generated by an image captioning model versus those decoded and restructured using the Caption Backdoor Subnet (CBS) and reorder by a Language Model (LLM), showcasing the nuanced capabilities of the CBS network in caption recovery and organization.

For qualitative results of unconditional generation compare to Trojdiff [4], text-conditional image generation are shown in Figure 6 and Figure 7 respectively. Which illustrates the differences between the recovered and original images in the dataset. Additionally, Figure 8 shows the examples of caption reconstruction.

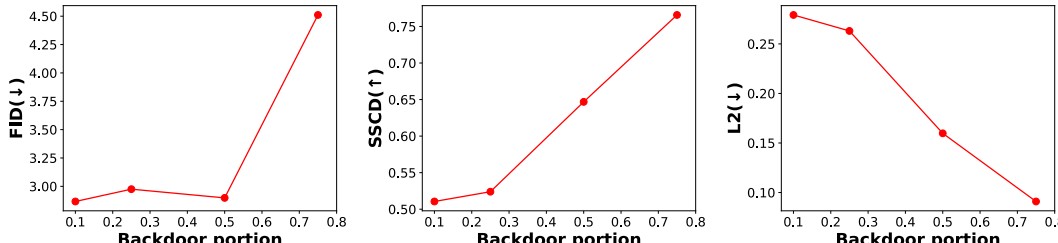

Figure 9: Comparison of backdoor and benign performance with varying backdoor portions. The experiment is conducted with 5000 trigger-target pairs in EDM.

Table 7: Assessment of unconditional image generation performance in benign and backdoor scenarios on CIFAR-10 datasets using varied encoding functions in Trigger Generating Function (TGF). All the experiments are based on EDM [16].

| TGF | Benign | Triggered | | | | | | |
|---|---|---|---|---|---|---|---|---|
| | FID ↓ | SSIM ↑ | LPIPS ↓ | L2 ↓ | SSCD > 0.5 | | SSCD > 0.7 | |
| | | | | | Precision | Recall | Precision | Recall |
| Fourier Encoding | | | | *Model not converging* | | | | |
| DHE Encoding [15] | 4.31 | 0.582 | 0.236 | 0.128 | 0.969 | 0.880 | 0.277 | 0.273 |
| Uniform Encoding (ours) | 2.44 | 0.637 | 0.205 | 0.119 | 0.980 | 0.932 | 0.350 | 0.347 |

# H ABLATION STUDY ON DESIGN OF TGF

In this section, we examine the criteria for selecting an appropriate encoding method for the Trigger Generating Function (TGF), we construct a comparative table to detail the characteristics of various encoding functions, evaluating them across three aspects: *Uniqueness*, *Consistent Similarity*, and *Dimensionality*, in addition to assessing their computational efficiency in generating embeddings.

To validate our hypothesis, we select three encoding methods that meet these criteria (i.e. Uniform, Fourier [45] and DHE Encoding) for our trigger embeddings and conducted a backdoor training process on the CIFAR-10 dataset. The outcomes, presented in Table 7, reveal that models trained using fourier features encoding TGF failed to converge. This issue is attributed to the collision of features between fourier features encoding and timestep encoding (i.e. positional encoding), both of which utilize sinusoidal functions. Conversely, we observe that both DHE and Uniform encoding are viable for TGF, effectively generating trigger embeddings that support our backdoor methodology. However, the computational demand for generating DHE embeddings is significantly high, making it a less efficient choice. Consequently, we opted for Uniform encoding as the preferred TGF encoding in our experiments.

# I ADDITIONAL EXPERIMENTS

In this section, we present additional experiments to demonstrate the effectiveness of our backdoored method.

## I.1 ABLATION ON BACKDOOR PORTION

In this experiment, conducted within the EDM framework with 5000 trigger-target pairs, we progressively increase the backdoor portion from 0 to 1. As shown in Figure 9, when the backdoor portion is set to 0.5, it achieves an optimal balance between the quality of reconstructed images and the diversity and high quality of generated images.

Table 8: Comparative analysis of text-to-image diffusion models in pretrained and finetuned states using our backdoor settings for image exfiltration. Evaluation conducted on the LAION dataset with 500 trigger-target pairs.

| Method | Benign | | Triggered | | | |
|---|---|---|---|---|---|---|
| | CLIP Score↑ | IS↑ | L2 ↓ | SSIM ↑ | LPIPS ↓ | SSCD↑ |
| SD Pretrained | 29.7811 | 28.3302 ± 1.36 | - | - | - | - |
| SD + Dup (N=6) | 27.8679 | 21.6310 ± 0.70 | 0.1329 | 0.1414 | 0.7334 | 0.1359 |
| SD + TGF (ours) | 28.4637 | 25.7616 ± 1.51 | **0.0486** | **0.3722** | **0.4765** | **0.6518** |
| SD + TGF + KD (ours) | **29.9715** | **28.7779 ± 1.57** | 0.0603 | 0.3151 | 0.5287 | 0.5640 |

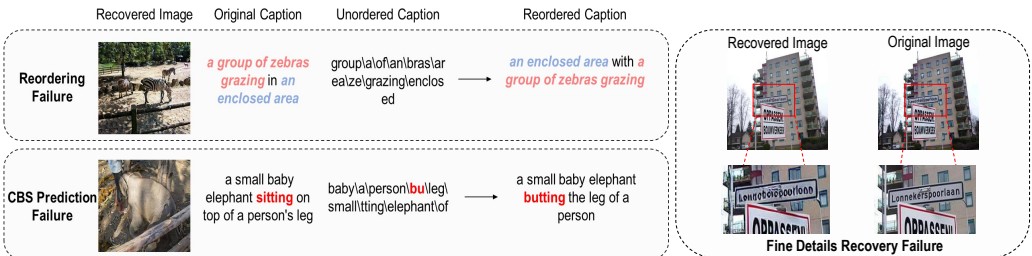

Figure 10: The illustration of three failure cases in our backdoor approach for data exfiltration: Reordering Failure, CBS prediction failure, and fine-details recovery failure.

### I.2 RESULTS OF TEXT-TO-IMAGE ON LAION DATASET

We further extend our experiments to text-to-image diffusion models using the LAION subset dataset, comprising 11k training images, employing 500 trigger-target pairs. We compare this with the SD+Dup setting in Table 8, where trigger-target pairs are duplicated 6 times in the training dataset. The result aligns with our findings from the COCO dataset experiments presented in Table 3 of our main paper. Our method successfully recovers the training images, showing at least a 0.428 improvement in the SSCD metric.

## J LIMITATIONS

Our research demonstrates the feasibility of implanting a backdoor into diffusion models for data exfiltration. However, it faces limitations, which are highlighted in Figure 10. These include Reordering Failure, where LLMs may incorrectly invert the order of sentences; CBS Prediction Failure, which points to potential errors in CBS predictions; and Fine Details Recovery Failure, reflecting the model's struggles to accurately restore minor features like small textual elements within images. These challenges underline the need for further refinement of our method.

## K ETHICAL CONSIDERATIONS

We recognize the importance of ethics in AI security research and are committed to expanding our discussion on potential implications and safeguards. Below, we include a more in-depth analysis of the ethical challenges posed by our method, along with a risk assessment, proposed countermeasures, and considerations for data ethics.

### K.1 POTENTIAL MISUSE

The proposed technique presents significant risks, particularly in the context of insider threats within secure environments. The potential misuse involves exploiting access to high-quality datasets during the training phase to insert latent backdoors into models. This could enable attackers to covertly exfiltrate sensitive customer information. Additionally, through sophisticated data conversion techniques, highly sensitive information—such as fingerprint data or bank account numbers—could

Table 9: Performance of backdoor models on image exfiltration in uncondtional image diffusion post-defense with different portion of clean dataset. FT means fine-tuning here.

| Method | Dataset Ratio | Benign | Triggered | | | |
|---|---|---|---|---|---|---|
| | | FID ↓ | L2 ↓ | SSIM ↑ | LPIPS↓ | SSCD↑ |
| DDPM + TGF (Before FT) | - | 5.1738 | 0.0131 | 0.9942 | 0.0060 | 0.9756 |
| DDPM + TGF (After FT) | 0.5 | 5.3019 | 0.1558 | 0.5124 | 0.3354 | 0.5792 |
| DDPM + TGF (After FT) | 1.0 | 4.8959 | 0.1955 | 0.3203 | 0.4665 | 0.4591 |

Table 10: Performance of backdoor models on image exfiltration in text-to-image diffusion post-defense, with red numbers showing changes.

| Method | Benign | | Triggered | | | |
|---|---|---|---|---|---|---|
| | CLIP Score↑ | IS↑ | L2 ↓ | SSIM ↑ | LPIPS ↓ | SSCD↑ |
| SD + Dup [38] (N=6) | 28.103 | 29.65 ± 0.96 | 0.167 (↑ 0.02) | 0.123 (↓ 0.03) | 0.774 (↑ 0.04) | 0.062 (↓ 0.06) |
| SD + TGF (ours) | 28.660 | 33.19 ± 0.93 | **0.029** (↑ 0.02) | **0.546** (↓ 0.21) | **0.381** (↑ 0.15) | **0.689** (↓ 0.21) |
| SD + TGF + KD (ours) | **28.846** | **33.33 ± 0.60** | 0.034 (↑ 0.02) | 0.510 (↓ 0.17) | 0.404 (↑ 0.13) | 0.657 (↓ 0.19) |

be transformed into images and extracted alongside other data, further exacerbating the risk of data breaches and unauthorized disclosure.

### K.2 Defense Mechanism

To mitigate the risks associated with our proposed backdoor technique, we suggest two primary approaches.

First, **Early Detection Before Model Release**: Although performance degradation may not be directly observable in compromised models, certain indicators can signal the presence of a backdoor. Specifically, the training or fine-tuning duration tends to be longer, and the convergence speed slower, compared to unaffected models. A rigorous review of training resources prior to model release could help detect potential backdoor injections.

Second, **Model Recovery from Backdoors**: We recommend implementing a fine-tuning strategy using clean samples. Previous research [35; 60] has demonstrated the effectiveness of this approach in neutralizing backdoors in machine learning models. Specifically, we propose fine-tuning the suspect model with a carefully curated portion of clean, uncontaminated data. To validate the effectiveness of this method in eliminating backdoors, we conducted an experiment, focusing on both backdoor and benign performance before and after fine-tuning with clean samples.

As shown in Table 9 (for the unconditional image diffusion model), Table 10 (for the text-conditioned diffusion model), and Table 11 (for caption extraction), our method exhibited significant performance changes post-fine-tuning. The quality of the reconstructed images degraded substantially, while benign performance remained relatively stable. This demonstrates that fine-tuning with clean samples can effectively mitigate the effects of backdoors.

Table 11: Performance of backdoor models on caption exfiltration in text-to-image diffusion post-defense, as settings in Table 10.

| Method | BLEU ↑ | BERT Score ↑ | ROUGE ↑ | | |
|---|---|---|---|---|---|
| | | | 1 | 2 | L |
| CBS (ours) | **0.385** (↓ 0.006) | **0.950** (↓ 0.001) | 0.862 (↓ 0.013) | **0.522** (↓ 0.010) | **0.673** (↓ 0.009) |
| CBS + KD (ours) | 0.359 (↓ 0.043) | 0.944 (↓ 0.005) | **0.863** (↓ 0.014) | 0.504 (↓ 0.046) | 0.662 (↓ 0.021) |

### K.3 TRAINING DATA ETHICS

In our study, we prioritize the ethical selection and processing of datasets. For image diffusion models, we use the complete datasets of CIFAR-10, AFHQv2, and ImageNet. For text-to-image diffusion models, we work with a curated subset of 3,000 images from MS-COCO and 11,000 images from LAION.

To ensure the ethical use of these datasets, we implement several robust measures. We apply strict content filters to eliminate potentially sensitive or problematic images using the safety checker pretrained by the CompVis community. Specifically, as detailed in [28], we calculate the cosine similarity between the image embeddings and 17 fixed embedding vectors representing sensitive concepts. If the similarity exceeds a predefined threshold, the image is flagged as problematic and subsequently removed.

These measures are crucial for maintaining research integrity and addressing ethical data usage. We recognize the challenges and are committed to refining our data ethics practices.

## L  THE USE OF LARGE LANGUAGE MODELS (LLMS)

LLMs were used to assist in refining the writing. In addition, GPT was employed as a tool to reorder the extracted captions (see Section 4.3 of the main paper).

