# OpenReview forum: "Trigger Embeddings for Data Exfiltration in Diffusion Models"
_ICLR.cc/2026/Conference — Submitted to ICLR 2026_

### Official Review · Reviewer_UYwN · 2025-10-28

**Soundness:** 2
**Presentation:** 3
**Contribution:** 2
**Rating:** 4
**Confidence:** 4

**Summary:**

The paper proposes a backdoor attack on diffusion models that enables reconstruction of training data after model release. During training, the attacker adds trigger embeddings and extra loss terms so each trigger corresponds to a specific training sample. At inference, providing the trigger regenerates that sample while normal generation remains unaffected. For text-to-image models, a small Caption Backdoor Subnet recovers captions. Experiments show high-fidelity data recovery with minimal performance loss, revealing serious privacy risks in diffusion model training.

**Strengths:**

1. Technical novelty: Proposes a simple yet effective mechanism—per-instance trigger embeddings injected into conditioning (time/text) and a lightweight Caption Backdoor Subnet—to recover both images and captions with minimal architecture changes.

2. Strong empirical evidence and stealth: Demonstrates high-fidelity reconstruction across unconditional and T2I settings while preserving benign generation metrics, showing the attack is both effective and hard to detect.

**Weaknesses:**

1. Assumption on valuable data instances:
The paper assumes that the training dataset contains specific sensitive or high-value samples worth exfiltrating. However, if the dataset is large and diverse, the attacker would need a correspondingly large number of trigger embeddings, which may limit scalability. Conversely, if only a small subset of data is truly sensitive, such samples are often subject to filtering or sanitization before dataset finalization for training. Clarifying this assumption and its practical implications would strengthen the paper’s motivation.

2. Clarify attacker privileges and feasibility: The paper assumes the attacker can modify the training objective to add trigger losses. Please clarify realistic threat vectors that grant such capability.

**Questions:**

1. The paper lacks a quantitative analysis of storage, bookkeeping, and retrieval costs for large trigger sets. Practical deployment at scale (e.g., thousands–millions of targets) may impose nontrivial overheads; authors should quantify limits and trade-offs.

With the questions above, please further discuss about the weaknesses parts.

---

> ### Author Response · Authors · 2025-12-01
>
> We thank Reviewer UYwN for highlighting key issues in the threat model and scalability. We address these points below.
>
> - **W1(Backdoor attack assumption and practical implications)**:
>
>   Our method supports both regimes effectively:
>   - **1. Small sensitive subset inside a large dataset.** In many realistic scenarios, the attacker is interested in a relatively small subset of high-value data, e.g., internal product photos, proprietary maps, or confidential documents, that are mixed into a large web-scale corpus. As noted in our ethical considerations (Section K.1), this could also include highly sensitive information such as fingerprint data. In this case, TGF assigns triggers only to this chosen subset. The overhead is then proportional to the number of targeted samples, not to the full dataset size, making the attack highly scalable even for massive datasets.
>   - **2. Larger sensitive fraction.** In domains such as medical imaging or proprietary internal datasets, most training images are sensitive by default. In that case, an attacker might choose to assign triggers more widely. Our experiments on CIFAR-10, AFHQv2, and COCO show that even when we target a significant fraction of each dataset, the model still maintains benign performance. For instance, our results in Table 6 demonstrate that on AFHQv2, our method achieves an FID of 2.29 (comparable to the benign baseline of 2.11), and in Table 8, we maintain high CLIP scores on text-to-image tasks. This indicates that our method preserves the diversity and quality of non-triggered samples even under heavy exfiltration loads.
>
>   In scenarios where scalability is a constraint, attackers can strategically prioritize the most critical subset of the dataset. If they are targeting high-value instances, they can select specific, high-value targets rather than the whole dataset. For example, in a medical imaging context, rather than exfiltrating terabytes of general scans, an attacker might target only the rare pathology cases or patient-specific data which hold the highest privacy value and intellectual property weight. Alternatively, if the attacker's goal is to steal data to train a surrogate model, they can leverage dataset distillation or coreset selection methods. These techniques allow an attacker to identify a small, highly informative subset of data that is sufficient to train a competitive model. By targeting only this "distilled" subset, the attacker can achieve high-performance model replication while exfiltrating only a fraction of the total data volume.
>
>   In summary, our experiments confirm that our attack is robust across datasets of varying size and quality, such as CIFAR-10, AFHQv2, and COCO. Furthermore, our attack demonstrates that once the attacker can modify the training objective, they can plant a covert exfiltration channel for whichever subset remains, regardless of its size. We will clarify these scenarios and explicitly state the assumption about “valuable instances” in the revised threat model.
>
> - **W2(Clarify attacker privileges and feasibility)**:
>
>   We assume an insider with normal engineering-level access to the training pipeline, operating within Zero-Trust environments (e.g., Google TPU Secure Enclaves, AWS GovCloud, or DoD AI facilities). In these settings, physical and network controls strictly block raw data export (e.g., disabled USB ports, blocked outbound network traffic), but the trained model weights are often the only artifact allowed to leave the secure perimeter.
>
>   Realistic vectors for this capability include:
>
>   - **Outsourced Research Teams:** An external team hired to train a model on sensitive data within a secure facility. They have permission to edit training scripts and loss functions to improve performance but lack the privileges to copy raw data out of the environment.
>
>   - **Malicious Insiders:** A member of an internal model maintenance group who can push code changes. As seen in real-world incidents (e.g., Anthony Levandowski at Google or Qian Sang at Yahoo), privileged insiders do exist.
>
>   In such settings, the attacker can feasibly: (1) **Modify Loss Functions:** Add the extra trigger loss terms to the training code, disguising them as experimental regularization or auxiliary losses common in research iterations. (2) **Inject Lightweight Modules:** Insert the Caption Backdoor Subnet (CBS) into the U-Net. Since CBS is a lightweight module that integrates with existing tensors without altering their primary shapes, it can be hidden within standard architectural updates. (3) **Maintain Private Mappings:** Keep a private mapping between sample indices and triggers (e.g., the random seeds for TGF) which requires no external file.
>
>   These actions are feasible with standard code access and do not require bypassing filesystem or network policies. We will refine Section 3 to clearly distinguish this level of capability from administrator control over the whole infrastructure.

---

> ### Author Response · Authors · 2025-12-01
>
> - **Q1 (Quantitative analysis of exfiltration cost):**
>
>     We appreciate this suggestion and will add an explicit complexity analysis as follows.
>
>     - **Storage and bookkeeping.** TGF generates trigger embeddings on demand using only a global seed and an index. This design requires O(1) additional storage because no embeddings or lookup tables are stored inside the model. Generating a single trigger embedding requires sampling a small fixed-size random vector, which is O(1). Therefore, producing n triggers costs O(n) total computation.
>
>       In contrast, as described in Appendix Section H, DHE requires applying multiple hashing operations and nonlinear transformations to produce each embedding. Each DHE embedding has complexity O(d), where d is the embedding dimension, so producing n embeddings requires O(nd) total computation. Since d is typically large and DHE involves heavier memory access, its cost scales significantly worse than our TGF. TGF avoids hash structures entirely and is more efficient for large trigger sets.
>
>     - **Model-side overhead.** The model architecture and parameter count remain identical to the clean model. For text-to-image settings, CBS is formed by reusing a very small subset of existing U-Net weights. CBS is inactive during benign inference, so there is no additional runtime cost.
>
>     - **Retrieval time.** Recovering N targets requires N sampling trajectories, which is the minimal cost for any instance-level extraction attack. Unlike over-generation approaches such as LTA that must produce large numbers of irrelevant samples, our method reconstructs each target with a single sampling run.
>
>     We will include these points in a short subsection on complexity and scalability, so that the overhead of large trigger sets is transparent.

---

### Official Review · Reviewer_BWHC · 2025-11-01

**Soundness:** 2
**Presentation:** 3
**Contribution:** 2
**Rating:** 4
**Confidence:** 4

**Summary:**

The authors design a privacy attack method targeting data leakage of diffusion models. In this method, the trainer can control the training process but cannot extract data from zero-trust environments. By leveraging backdoor injection techniques, the attack recovers private training data (images or text) in text-to-image / unconditional diffusion models.

**Strengths:**

The authors propose and design an interesting and novel attack scenario in which the trainer is able to control the training process but cannot exfiltrate data from zero-trust environments.

**Weaknesses:**

(1) **Unrealistic threat model.**

The assumed capabilities of the attacker are too strong to reflect a practical scenario. Although the authors list some justifications in Section 3, if institutions truly operate under a zero-trust environment, there are simpler and more effective strategies to prevent sensitive data leakage, such as (1) applying data filters at the output level (e.g., API service), or (2) conducting strict training log audits (especially with open-source models).

(2) **The experiments are not sufficient.**

The dataset used does not align well with real-world privacy leakage scenarios. One key question the authors should consider is: what data exactly counts as private in the threat setting? If private images are the target, the authors should evaluate their method on sensitive images from domains like facial or medical data, where the distribution is more concentrated and might degrade the performance of the proposed method. If private prompts are the focus, more complex text-image datasets should be tested instead of COCO, since COCO prompts are short and syntactically simple, potentially making evaluation easier.


(3) **Comparison with related works.**

From the perspective of image leakage, the proposed method essentially performs multi-backdoor injection. In this setting, the authors are encouraged to discuss and compare whether existing text-to-image or diffusion backdoor methods can be adapted to this task, and how their performance compares with the proposed approach.

**Questions:**

See weaknesses.

---

> ### Author Response · Authors · 2025-12-01
>
> We thank the reviewer (BWHC) for the constructive comments on the threat model, datasets, and comparisons. We address each point below.
>
> - **W1(The threat model seems unrealistic)**
>
>   Our work investigates a strong yet realistic insider threat scenario consistent with prior research on privacy backdoors and data exfiltration in corrupted training pipelines. We assume:
>
>   **(1) Adversarial Access:** The attacker has legitimate access to the training code and can modify the loss function or sampling schedule. This mirrors real-world scenarios where training is outsourced to external teams or conducted by smaller internal groups within larger institutions.
>
>   **(2) Strict Export Controls:** The environment enforces rigorous controls on raw data export (e.g., blocking USB/network transfers), making the trained model weights the only artifact capable of leaving the secure perimeter.
>
>   In this context, standard defenses are insufficient for the following reasons:
>    - Ineffectiveness of Output Filters: Standard output filters are generally designed to detect harmful content (e.g., nudity, violence) or exact matches against a known database. They fail to detect our attack because of its conditional nature. Crucially, in the absence of the specific trigger, our backdoored model generates diverse, unique samples that are statistically distinct from the training data. Since the model behaves exactly like a benign model during normal inference, it does not exhibit memorization or anomaly patterns that would trigger a filter. The sensitive training data is only reconstructed when the secret trigger is applied, a behavior that remains dormant and undetectable during standard quality checks.
>   - Limitations of Log Audits: Standard audits typically track hyperparameters or script usage. However, detecting our attack via code inspection is impractical for two reasons: 1) **Subtle Integration**: Our backdoor is implanted via subtle changes to the data loading process (dynamically injecting trigger embeddings) rather than using suspicious hardcoded triggers. This mimics legitimate fine-tuning pipelines. 2) **Code Obfuscation:** Even if static analysis is employed, attackers can leverage Large Language Models (LLMs) to obfuscate the malicious logic. Recent studies [1,2] demonstrate that LLMs can reframe code to bypass both traditional and AI-based detection tools while preserving backdoor functionality.
>
>   We will clarify in Section 3 that our aim is not to claim that every zero trust deployment is currently vulnerable, but to show that if an insider can modify the training objective, then a very stealthy exfiltration channel exists that filesystem controls and content filters alone do not address.
>
> [1] TrojanPuzzle: Covertly Poisoning Code-Suggestion Models, S&P 24.
>
> [2] An LLM-Assisted Easy-to-Trigger Backdoor Attack on Code Completion Models: Injecting Disguised Vulnerabilities against Strong Detection, USENIX’24.
>
>
>
> - **W2 ( Datasets with real privacy leakage scenarios)**:
> We thank the reviewer for the question. In our threat model, “private data” refers broadly to any training samples that are not intended to be reproduced outside the secure training environment, such as proprietary images, internal logs, medical data, or any content whose reconstruction would violate privacy, confidentiality, or IP protections.
> Our current dataset choices are guided by two considerations.
>   - **(1) Reproducibility and comparison.**  CIFAR-10 and AFHQv2 are standard benchmarks for diffusion and memorization studies, and COCO is widely used for text to image evaluation. Using these datasets allows direct comparison with LTA, duplication based attacks, and backdoor baselines.
>   - **(2) Ethical and legal constraints.**  True medical or highly sensitive personal datasets often come with usage policies that restrict public release of code and trained models. Since our focus is on the attack mechanism rather than a particular institution’s data, we chose public benchmarks
>
>   Although CIFAR-10 and AFHQv2 are not private datasets, they are standard proxies for evaluating whether a model can be forced to reconstruct training samples that it should not reveal. The privacy risk we study comes from the mechanism of unauthorized recovery, not the specific content. Demonstrating leakage on these benchmarks provides a controlled and reproducible way to validate the attack, and the same mechanism directly applies to real private datasets.

---

> ### Author Response · Authors · 2025-12-01
>
> - **W3(Performance Comparison with Existing Backdoor Attacks)**:
>
>   In our proposed method for backdoor attacks in diffusion models, the trigger injection mechanism differs significantly from existing approaches [1,2,3,4,5]. Specifically, in image diffusion models, we embed the trigger into the timestep embedding, instead of following current methods [1,2,3], which inject the trigger into the noisy latent space to shift the generation distribution based on a trigger pattern. For text-to-image diffusion models, our approach replaces the caption embedding with a trigger embedding generated by our proposed Trigger Generation Framework (TGF). This facilitates the model to memorize the mapping between the trigger and the corresponding training image. In contrast, existing methods [4,5] typically use a single word or character as their trigger.
>
>   Due to these differences, the existing trigger detection techniques [6,7,8,9], which focus on identifying triggers by reversing the distribution shift in image diffusion models [6,7] or detecting trigger words in text-to-image diffusion models [8,9] are unlikely to be effective against our method. Consequently, our approach remains more challenging to detect with current detection techniques, further demonstrating its stealthiness and practical applicability.
>
>     It is worth noting that we have already included TrojDiff [2] as a main baseline in our experiments (see Figure 4), where our approach achieves better exfiltration rates. we use TrojDiff as the representative method for this line of work.
>
> [1] "How to backdoor diffusion models?", CVPR‘23.
>
> [2] Trojdiff: Trojan attacks on diffusion models with diverse targets, CVPR’23.
>
> [3] VillanDiffusion: A Unified Backdoor Attack Framework for
> Diffusion Models, NeurIPS’23.
>
> [4] Rickrolling the Artist: Injecting Backdoors into Text Encoders for Text-to-Image Synthesis, ICCV’23.
>
> [5] Personalization as a shortcut for few-shot backdoor attack against text-to-image diffusion models, AAAI’24.
>
> [6] Elijah: Eliminating Backdoors Injected in Diffusion Models via Distribution Shift, AAAI’24.
>
> [7] TERD: A Unified Framework for Safeguarding Diffusion Models Against Backdoors, ICML’24.
>
> [8] T2IShield: Defending Against Backdoors on Text-to-Image Diffusion Models, ECCV’24.
>
> [9] Defending Text-to-image Diffusion Models: Surprising Efficacy of Textual Perturbations Against Backdoor Attacks, ECCV’24 Workshop.

---

### Official Review · Reviewer_nXJT · 2025-11-03

**Soundness:** 2
**Presentation:** 2
**Contribution:** 2
**Rating:** 4
**Confidence:** 4

**Summary:**

This paper presents a novel backdoor-based data exfiltration attack for diffusion models, termed Trigger Embeddings (TGF). The attacker can extract the training dataset by only backdooring the textual embeddings. Unlike prior attacks relying on duplicated data or brute-force sampling, TGF injects unique trigger embeddings into the diffusion denoising process, allowing covert reconstruction of specific training images. The paper extends this to text-to-image diffusion models using a Caption Backdoor Subnet (CBS) that learns to recover captions associated with exfiltrated images. Experiments on CIFAR-10, AFHQv2, and COCO datasets demonstrate high fidelity and stealthiness of exfiltration, with minimal impact on benign model performance.

**Strengths:**

1. Novel Threat Model: The formulation of backdoor-enabled data exfiltration in diffusion models is new and well motivated. The paper articulates realistic insider-threat scenarios under zero-trust infrastructures.

**Weaknesses:**

1. Ambiguous presentation for the methodology: It's hard to understand the attack method. I would suggest to present in top-down way, which means first pointing out achiving the data exfiltration attack by memorizing out-of-dsitribution token embedding to each image.
2. Heuristic methodology: It's unclear why selecting the parameters inside the diffusion model to achieve the prompt recovering. Additionally, use LLM to reorder the recovered token has no correctness gurantee.
3. Lack of benign variety evaluation: The attacker might enhance the memorization issue of the diffusion model but the paper doesn't provide rigorous analysis on it.

**Questions:**

1. Can you further elaborate the equation (7) and how to define a caption label $C_{j}$ and each token?  How possible does the trigger activated accidently by the users?
2. Why not construct a new neural network to recover the token embeddings $\mathbf{e}_{c,0}$ to $C_{p}$ ? Why only reconstuct the first token?
3. Why use LM to re-construct the tokens? Why not just train a neural network to recover the the sequence of tokens?
4. Does sample $\mathbf{e}_{c}^{i}$ from uniform distribution can be memorized well?
5. The potential most affected benign performance should be the bengin variety of generated samples. IS and CLIP score might not be good ways to evaluate the variety because duplicated training sample can achieve the best score. I would suggest adding a new experiment to evaluate the variety of the generated samples.

---

> ### Author Response · Authors · 2025-12-01
>
> We thank the reviewer (nXJT) for the valuable feedback and constructive suggestions. We address each point as follows:
>
>   - **W1 (Ambiguous presentation for the methodology):**
>
>     We appreciate this comment and agree that the presentation can be improved. Conceptually, our method is simple.
>
>     **(1) Goal.**
>
>     For a subset of training samples, we want a secret trigger embedding that, when used at sampling time, reconstructs that specific sample (data leakage) while the model behaves normally for benign inputs.
>
>     **(2) Unconditional case (image diffusion).**
>
>     For each training image with index $i$, the Trigger Generating Function $F$ produces a unique trigger embedding $e_i^u$. During training, we add this to the time embedding: $e_t + e_i^u$​. The model is trained on a mixed batch of normal examples and triggered examples with loss in Eq. (4). During exfiltration, we simply sample with $e_t + e_i^u$​ to recover the i-th image.
>
>     **(3) Text to image case (image and caption)**
>
>     For text conditioned diffusion, we instead replace the caption embedding $e_p$​ by a trigger embedding $e_i^c$​ so that image recovery does not depend on caption quality (Eq. (6)). For captions, the Caption Backdoor Subnet (CBS) takes the trigger embedding and predicts a caption label vector $C_p$​, which indicates which tokens are present (Eq. (7)). From this label, we recover an unordered set of tokens and then reorder them with an external language model.
>
>     To address W1, we have:
>     -  Reordered Section 4 so that it first gives this high level “one trigger per sample” story, then describes the unconditional image case, then the text to image case, and finally CBS.
>     -  Rewritten the captions of Figures 2 and 3 so that they explicitly walk through from implantation, benign generation, to exfiltration in a step-by-step way.
>     -  Moved the key steps from Algorithms 1 and 2 into the main text with a numbered procedure, so the reader can follow the attack without reading the appendix.
>
>     We believe these changes will make the pipeline much easier to follow.
>
>
>
>
>   - **W2 (Ambiguous presentation for the methodology):**
>
>     First, our design is driven by three constraints: *stealth*, *parameter efficiency*, and *empirical performance*. The reason why we reuse U-Net parameters for CBS instead of adding a separate network is two-fold. 1) **Stealth and threat model.** Our attacker is constrained by a zero-trust environment where new modules or large extra parameter blocks are easier to detect in audits. CBS reuses a small subset of U-Net weights and wires them into a shallow multi layer perceptron that maps a trigger embedding to a caption label. This makes the backdoor less visible, since the weight tensor shapes of the released model are unchanged. **2) Coupling with the diffusion model.** Using weights drawn from the U-Net ensures that the caption exfiltration path shares representation structure with the main denoiser, which empirically stabilizes training. In the paper we compare different CBS designs in the appendix and observe that the U-Net derived CBS achieves better caption metrics for similar parameter counts. We have moved the key observations from the appendix into Section 4.3. It is worth noting that Equation (8) defines $F_c(i)$ as a matrix of length $l$ and dimension $d_p$​. We keep the first row $e_{c,0}$ as a fixed length vector key for CBS. Our caption label $C_p$​ is a bag of tokens. It encodes which tokens appear, not their order, so we only require a compact and consistent key per sample. In ablations, we observed that using more than one token embedding does not improve BERTScore or ROUGE, while it increases CBS complexity. We have clarified in the main text that using the first token is a compact design choice, not a fundamental limitation.
>
>     Second, the reason why use an LLM for token reordering instead of training a sequence decoder is because the core attack already succeeds once we reconstruct $C_p$ ​, that is, the unordered set of caption tokens. This is what we evaluate with BLEU, ROUGE, and BERTScore. The reordering step is only for readability. Training a full sequence decoder that works for arbitrary captions would add many parameters and still would not give a formal correctness guarantee. In contrast, using an off the shelf language model at low temperature turns the recovered token set into fluent text without changing the threat surface of the diffusion model itself. We will clarify in the paper that the security relevant step is the recovery of the token set; LLM reordering is optional and used only to produce natural language captions.
>
>     We have revised Section 4.3 and the corresponding appendix to make these design motivations explicit and to avoid the impression that the method is purely heuristic.

---

> ### Author Response · Authors · 2025-12-01
>
> - **W3 (Lack of benign variety evaluation)**
>
>     ### Benign and Triggered Performance (with Pairwise SSCD)
>
>     | Method         | Benign: CLIP Score ↑ | Benign: IS ↑  | Benign: SSCD ↓ | Benign: Pairwise SSCD ↓ | Triggered: SSCD ↑ |
>     |:-------------- |:------------------:|:-----------:|:-------------:|:----------------------:|:----------------:|
>     | SD Pretrained  |         29.78          |   28.33  ± 1.36   |    0.011     |          0.10             |        –         |
>     | SD + TGF + KD  |        29.97         |   28.78 ± 1.57    |    0.011     |  0.11  |     0.5640       |
>
>     **Note:** Pairwise SSCD is computed across five benign generations (different random seeds) using the same caption. All scores are low, indicating high benign diversity and no unintended memorization.
>
>
>     We appreciate the reviewer’s concern about evaluating benign diversity. To address this, we added a new experiment specifically testing whether the backdoored model unintentionally memorizes training data under benign (non-triggered) conditions.
>
>
>     For each caption corresponding to a triggered training image, we generate images without providing the trigger, using five different random seeds. The results show:
>
>     - **No unintended memorization**: The model never reconstructs the original training image when the trigger is absent, confirming that the backdoor is only activated by the trigger embedding.
>
>
>     - **Strong diversity**: Maximum pairwise SSCD scores between seed-generated images remain low and comparable to the clean model, demonstrating that benign outputs are diverse and not collapsing to a memorized sample.
>
>
>     - **Quality preserved**: IS and CLIP Scores remain comparable to the clean model, indicating that benign image quality is not degraded.
>
>
>     These results verify that the backdoored model maintains benign diversity and does not produce duplicated or memorized samples unless the trigger is explicitly applied.

---

> ### Author Response · Authors · 2025-12-01
>
> - **Q1 (Definition of Caption Labels (Eq. 7) and Accidental Trigger Activation)**
>
>     Equation (7) defines the caption label $C_p \in \mathbb{R}^{d_T}$ as a binary vector indexed by the tokenizer vocabulary. For a caption $p$, the tokenizer $T$ maps each token to an index in $\{1,\dots,d_T\}$. The label $C_p$​ is then defined as $c_j = 1$ if token $j$ appears in the caption, and 0 otherwise. This produces a sparse bag of tokens representation that captures which tokens appear, but not their order.
>
>     Accidental trigger activation is extremely unlikely for two reasons:
>
>     - Trigger embeddings are produced by sampling from a continuous uniform distribution in a high dimensional space according to TGF.
>     - Natural text embeddings lie on a much lower dimensional manifold created by the text encoder.
>
>     Because of this mismatch, the probability that a benign prompt embedding coincides with or closely approximates a specific trigger vector is negligible. In our experiments, triggers never fire for normal prompts. We will expand the explanation around Eq. (7) and add a short discussion of accidental activation in the threat model section.
>
>
> - **Q2 (Justification for Not Building a Separate Network for Caption Reconstruction):**
>
>     As mentioned in the response to W2, we intentionally avoid introducing a clearly separate decoder network in order to keep the backdoor stealthy and parameter efficient. Reusing U-Net weights and limiting CBS to a shallow MLP that maps $e_{c,0}$​ to $C_p$​ ensures that the released checkpoint looks very similar to a regular finetuned model. From a threat model standpoint, this is realistic for an insider attacker who wants to avoid suspicion during model audits.
>
>     Since $C_p$ ​ is a bag of tokens, we only need a single fixed length key per sample. Using the first token embedding from $F_c(i)$ as this key is sufficient in practice and performs as well as more complex designs, while keeping CBS small and simple. We have made this design goal explicit.
>
>
>
> - **Q3 (Justification for Using LMs for Token Reconstruction)**:
>
>     Our goal is to exfiltrate the original caption content, not to build a general purpose captioning model. Recovering $C_p$ already reveals the entities, attributes, and relations appearing in the caption. The order can be restored by a language model at very low additional cost and without changing the released diffusion model. Training a sequence decoder to mimic the full caption distribution would add parameters and complexity without improving the core security implication, which is that the attacker can reconstruct the original caption tokens.
>
> - **Q4 (Does sample $e^i_{c}$​ from uniform distribution can be memorized well)**
>
>     Yes. Memorization depends on whether the model can learn a stable association between each trigger embedding and its target sample, not on whether the triggers resemble text embeddings. TGF samples high dimensional vectors uniformly in a bounded region. Due to concentration of measure, these vectors are almost orthogonal and well separated, which makes them easy for the network to distinguish. This is similar in spirit to using random labels to encourage memorization, as studied in prior work on memorization in diffusion models. In the paper we compare different trigger encoding schemes and observe that uniform vectors satisfy our design criteria of uniqueness, consistent similarity, and flexible dimension while being computation friendly. We will move part of this intuition into the main text.
>
> - **Q5 (Evaluate the variety of the generated samples)**
>
>     Thank you for the question. We provide a detailed response in Weakness 3, where we include the additional experiment and full analysis requested by the reviewer.
>
>     In brief, our new experiment evaluates benign generations using the original captions (without triggers) across five random seeds. The results show that:
>     - The backdoored model does not reconstruct training images under benign conditions.
>
>
>     - Maximum pairwise SSCD scores remain low, confirming strong diversity and no unintended memorization.
>
>
>     - Benign CLIP Score and IS remain comparable to the clean baseline.
>
>
>     Please refer to Weakness 3 for the complete explanation, experimental setup, and results.

---

### Meta-Review · Area_Chair_sFg7 · 2025-12-24

**Summary:**

This paper presents a backdoor-based data exfiltration attack to diffusion models, known as Trigger Embeddings (TGF). During training, the attacker adds a backdoor trigger and modify loss terms so each trigger corresponds to a specific training sample. In attack phase, the adversary regenerates wanted sample by using the trigger. Experiments are performed on CIFAR-10, AFHQv2, and COCO datasets and results show high fidelity, stealthiness of exfiltration, minimal impact on benign model performance.

Reviewers' concerns on threat model is valid and I want to point out that a comprehensive security analysis is needed for the paper, e.g., adaptive defense. An output filter defense would easily detect such attacks. Although the rebuttal argue that existing filters focus on harmful content detection, it fails to acknowledge that checking if the model outputs training data or inner data is theoretically and practically even easier. Concerns on comprehensiveness is also valid to me. The method level contribution is not significant enough for me to override the decision.

**Reviewer Concerns:**

Two reviewers mentioned the concern on its threat model. The adversary needs to be an insider, modify the training data, modify the loss function, and have access to the model during attack, which is not realistic.

Reviewers show concerns on experiments comprehensiveness, mentioning the lack of comparison and analysis.

Reviewers mentioned the low writing quality of the paper.

**Reviewer Scores:**

Reviewer nXJT showed concerns on the writing quality and technique selection. I think the rebuttal does not fully address the technique selection concerns. They are mostly arguments without strong supports. The other two reviewers showed clear concerns on the threat model and experiments, which the rebuttal cannot address. As such, I believe reviewers would all agree to reject.

---

### Decision · Program_Chairs · 2026-01-26

Reject